# Fungicide Application Affects Nitrogen Utilization Efficiency, Grain Yield, and Quality of Winter Wheat

Ahossi Patrice Koua [1,†] , Mirza Majid Baig [1,†], Benedict Chijioke Oyiga [1], Jens Léon [1,2] and Agim Ballvora [1,*]

1 Institute of Crop Science and Resource Conservation, Plant Breeding, Faculty of Agriculture, Rheinische Friedrich-Wilhelms-University, Katzenburgweg 5, 53115 Bonn, Germany; koua_patrick@uni-bonn.de (A.P.K.); majidsolo2003@hotmail.com (M.M.B.); benedictoyiga@gmail.com (B.C.O.); ulp201@uni-bonn.de (J.L.)
2 Field Lab Campus Klein-Altendorf, University of Bonn, Klein-Altendorf 2, 53359 Rheinbach, Germany
* Correspondence: ballvora@uni-bonn.de; Tel.: +49-(0)228-73-7400; Fax: +49-(0)228-73-2045
† Both authors contributed equally to the manuscript.

**Abstract:** Nitrogen (N) is a vital component of crop production. Wheat yield varies significantly under different soil available N. Knowing how wheat responds to or interacts with N to produce grains is essential in the selection of N use efficient cultivars. We assessed in this study variations among wheat genotypes for productivity-related traits under three cropping systems (CS), high-nitrogen with fungicide (HN-WF), high-nitrogen without fungicide (HN-NF) and low-nitrogen without fungicide (LN-NF) in the 2015, 2016 and 2017 seasons. ANOVA results showed genotypes, CS, and their interactions significantly affected agronomic traits. Grain yield (GY) increased with higher leaf chlorophyll content, importantly under CS without N and fungicide supply. Yellow rust disease reduced the GY by 20% and 28% in 2015 and 2016, respectively. Moreover, averaged over growing seasons, GY was increased by 23.78% under CS with N supply, while it was greatly increased, by 52.84%, under CS with both N and fungicide application, indicating a synergistic effect of N and fungicide on GY. Fungicide supply greatly improved the crop ability to accumulate N during grain filling, and hence the grain protein content. Recently released cultivars outperformed the older ones in most agronomic traits including GY. Genotype performance and stability analysis for GY production showed differences in their stability levels under the three CS. The synergistic effect of nitrogen and fungicide on grain yield (GY) and the differences in yield stability levels of recently released wheat cultivars across three CS found in this study suggest that resource use efficiency can be improved via cultivar selection for targeted CS.

**Keywords:** nitrogen fertilization; fungicide treatment; cropping systems; yield components; yield performance; yield stability

## 1. Introduction

Wheat is one of the world's most important staple food crops [1] and plays a major role in global food security. Thus, improving wheat grain yield is essential towards feeding the world's growing population [2]. Nitrogen fertilizer is a key factor in the determination of cropping systems (CS), and has a strong effect on plant metabolism and biological processes that regulate plant growth and development [3,4]. It also affects wheat grain quality including gluten, protein and starch contents [5].

Considering the importance of nitrogen (N) in ensuring higher crop yield and productivity, farmers tend to overuse it [6,7], but pay little attention to important aspects like plant N uptake and utilization. An excessive application of nitrogen may lead to over-stimulation of tillering and plant vegetation (i.e., haying-off) that locks up carbohydrates in structural materials rather than transporting them to the storage organs for later use at the grain filling stage. Effectively, only one third of the applied N is utilized by plants for grain production, resulting in a huge waste of resources that harms the environment [8,9].

Buckwell and Nadeu [10] reported that, based on estimations between 2004 and 2011, apart from the northeast regions of Europe and mountain areas, most of the European Union (EU) is characterized by surpluses of nitrogen in agricultural land with an average of 49 to 80 kg·ha$^{-1}$.

Wheat cultivation and productivity are limited by diverse biotic and abiotic stress factors such as soil water content, soil and air temperatures and disease occurrence [11]. A direct relationship has been established between the N application and the incidence of yellow rust (*Puccinia striiformis* f. sp. *tritici*) [12,13], indicating that application of fungicides would decrease pathogen-related yield losses, especially under CS with higher amounts of fertilizers [14]. Improvement of wheat yield requires development of resilient CS [15], including selection of genetic resources that significantly increase plant productivity under limited and optimal nitrogen and agrochemical inputs. In the context of political and environmental constraints on agrochemical inputs and climatic changes, the reduction of agricultural inputs will contribute to reducing the negative impacts of agriculture on the environment [3]. Therefore, breeding to increase crops nitrogen use efficiency (NUE) in different crop management systems will assure high productivity along with lower economical costs and environmental threat [2].

A better understanding of the relationships existing among wheat productivity related traits, N fertilization and fungicides application would facilitate the identification and selection of high yielding cultivars for different targeted N application rates and CS. Several studies have revealed key components of NUE, including N uptake and N use efficiency [4,8]. They indicated that genetic variation exists among cultivars for traits related to NUE and recommended the use of broader germplasm including new and old varieties, landraces and wild relatives to gain insights into NUE in crops. The breeding progress in wheat has resulted in increasing yield under less optimum conditions via the identification and conservation of favorable genetic factors and haplotypes involved in stress adaptation in crops while eliminating detrimental genetic variants [16]. Ho, it has been also reported that newer released wheat cultivars perform poorly under less optimal conditions compared to the older released wheat cultivars [17]. To date, few studies have investigated the main and cumulative effects of fungicide and nitrogen on wheat productivity as well as their interactions under different CS at vegetative and reproductive stages. In the context of breeding for low and high input CS, sufficient research information needs to be provided to wheat growers to improve productivity under both conditions. In this study, 220 winter wheat cultivars released from the last 50 years with variable phenotypic traits were grown in three CS including one conventionally managed system with high nitrogen, fungicides and growth hormones. The aims of this study were (1) to explore the genetic variability existing among 220 cultivars for agronomic traits, disease response, and seed grain quality under effect of nitrogen and fungicide CS; (2) identify the GY most contributing traits among its components under each CS and (3) provide information on the breeding progress made in important traits to assure high productivity under less and high input conditions.

## 2. Materials and Methods

### 2.1. Plant Material, Experimental Design and Treatment

A total of 220 winter wheat cultivars described by Voss-Fels et al. [16] were used in this study. They were grown in the field at the Campus Klein-Altendorf, University of Bonn (50.61° N, 6.99° E, and 187 m above sea level) in the 2014/2015, 2015/2016 and 2016/2017 growing seasons. The experiments were performed in an alpha design consisting of 1320 plots under three CS with two replications each, using plot-in-plot systems. The CS treatments adopted were: LN-NF (no N fertilization, without fungicide), HN-NF (semi-intensive system with 220 kgN·ha$^{-1}$ mineral fertilizer adjusted for soil mineral nitrogen (Nmin), plant growth regulators, and no fungicide application), and HN-WF (intensive system with 220 kgN·ha$^{-1}$ adjusted for soil Nmin, plant growth regulators plus fungicide application). The soil information, including Nmin of the experimental site, the amount

of N fertilizer applied, and the agro-chemical input including fungicide are provided in Supplementary Table S1, Tables S2 and S3, respectively. Each plot was 6 m long and 2.5 m wide with a sowing density of 330 viable seeds per m$^2$ in rows spaced by 10.4 cm. The harvested plot was 5.0 m long and 1.65 m wide. The weather conditions during the experimental periods are summarized in Figure S1.

### 2.2. Soil Sampling and Fertilization

After seed sowing, soil samples were taken from 0 to 90 cm depth in 30 cm increments and analyzed for the available Nmin in the soil. Soil Nmin was determined by micro-Kjeldhal digestion method [18]. Ammonium N (NH$_4$$^+$N) was extracted by 2 M KCl and analyzed by using phenate method [19]. Nitrate N (NO$_3$$^-$N) was extracted by 1g/100 mL CaSO$_4$ and analyzed by phenol disulphonic acid method [18]. The soil characteristics of the experimental site are shown in Table S1.

### 2.3. Phenotypic Evaluation

The traits evaluated include: agronomic (plant height (PH), heading date (HD), spikes per m$^2$ (SNms), kernels per spikes (KNSp), kernels per m$^2$ (KNms), thousand kernels weight (TKW), harvest index (HI), plant biomass per m$^2$ weight (PBWms), and GY); physiological (chlorophyll content (SPAD)); disease response (yellow rust visual score (YR)), and grain quality (grain crude protein (GPC), grain starch content (GSC) and sedimentation). The heading date (HD) indicated the duration of vegetative period from germination until heading growth stage at BBCH59 (Biologische Bundesanstalt, Bundessortenamt und CHemische Industrie [20]). The severity of yellow rust was scored in the 2015 and 2016 growing seasons under all three cropping systems according to Pask et al. [21]. The full description of evaluated traits is given in Table S4.

### 2.4. Statistical Analyses of the Evaluated Traits

An analysis of variance (ANOVA) for all the traits was performed with a mixed-linear-model to determine the effect of CS, genotypes (cultivars), and their interactions across the three growing seasons. Restricted maximum-likelihood (REML) was adopted to estimate the variance parameters, and the best linear unbiased estimate (BLUEs) of all traits for each cultivar under different CS were generated. The resulted BLUEs were used for the subsequent downstream analyses. BLUEs of three CS for each trait were compared using Tukey's honestly significant differences (HSD) test to obtain significance groups [22]. A three ways analysis of variance was carried out, especially for GY and GNY, to estimate the existence of variation among genotypes, CS, years, and their interaction effects using ANOVA Procedure in SAS software [23].

A mixed-linear model with restricted maximum-likelihood (REML) was used to estimate the variance components due to genotypes ($\sigma^2_g$), CS ($\sigma^2_e$), and their interaction G $\times$ CS ($\sigma_{ge}^2$). These components were set as random effects in the model [23]. Thereafter, the broad-sense heritability (H$^2$) for all traits across growing seasons were calculated as described by Piepho and Möhring [24]) using the equation

$$H^2 = \sigma_g{}^2 / \sigma_p{}^2 \tag{1}$$

$$\text{with } \sigma_p{}^2 = \sigma_g{}^2 + \sigma_{ge}{}^2/m + \sigma^2/rm$$

where $\sigma_p{}^2$ is the phenotypic variance, *m* the number of studied CS, *r* the number of replicates per CS and $\sigma^2$ the residual error variance.

Pearson correlation analysis of genotypic means was performed using the package *Performance Analytics* in R [25] to assess the correlation between evaluated traits. We tested the significant difference among CS correlation coefficients of GY and its components through the r.test function for two independent correlations in a Fisher's z-test in the*psych* R package. Thereafter, the relationships between GY and traits of interest were evaluated using linear regression to quantify the contribution of the trait to GY. The regressions

were conducted using the lm function as implemented in R and path models analysis using *lavaan* and *semPlot* packages as described by Rosseel [26]. The differences among regression coefficients, namely slopes and intercepts of the three CS, were tested using linear regression models which included CS as categorical variables.

### 2.5. Effects of Nitrogen and Fungicide on the Evaluated Traits

The effects of nitrogen and fungicide on the evaluated traits were calculated with traits average values under each CS using the following formula.

$$N_{eff} = [P_{(HN-NF)} - P_{(LN-NF)}]/P_{(LN-NF)} \tag{2}$$

$$NF_{eff} = [P_{(HN-WF)} - P_{(LN-NF)}]/P_{(LN-NF)} \tag{3}$$

$$F_{eff} = [P_{(HN-WF)} - P_{(HN-NF)}]/P_{(HN-NF)} \tag{4}$$

where: $N_{eff}$, represents the nitrogen effect; $NF_{eff}$, the combined nitrogen plus fungicide effect and $F_{eff}$ the fungicide effect under high nitrogen CS.

Two indicators including NUE (nitrogen use efficiency) and NAE (nitrogen agronomy efficiency) were estimated according to Ma et al. (2019) and used to determine the N requirements for GY production. NUE and NAE were calculated as:

$$NUE \left(\frac{kg}{kg}\right) = Y_{Nav}/N_{av} \tag{5}$$

$$NAE \left(\frac{kg}{kg}\right) = (Y_N - Y_0)/A_N \tag{6}$$

where $Y_{Nav}$ (kg) is the GY harvested under respective CS, $N_{av}$ (kg) is the available nitrogen (Fertilizer and Nmin) in respective CS, $Y_N$ (kg) indicates the GY under high N, $Y_0$ (kg) is the GY obtained under low N and $A_N$ is the amount of applied N fertilizer under high N.

The effect of fungicide on NAE under HN-NF and HN-WF was investigated to evaluate how fungicide application could improve N use. We defined four classes of NAE depending on the amount of N used for GY production. The cultivar with the highest GY under HN-NF or HN-WF was considered as having converted 100% of the available nitrogen into GY, hence had the highest NAE ($NAE_{max}$). A cultivar *i* was class one (class1) when k = $NAE_i/NAE_{max} \times 100$ was less than 25%, class2 when k was more than 25% and less than 50%, class3 when k was between 50% and 75%, and with k greater than 75% was assigned as class4.

### 2.6. Estimation of the Breeding Progress Using the Wheat Diversity Panel

The breeding progress in the winter wheat panel for each trait was investigated with 209 cultivars whose year of release was known by the linear regression function. The BLUEs values of each cultivar averaged over three years growing seasons were used in the regression analysis. The absolute breeding progress (increase per year) was the slope of the linear regression line between the year of release and the trait of interest, as described by Lichthardt et al. [27].

### 2.7. GY Performance and Stability Analysis

To measure the GY performance of each cultivar under three CS across three years, the cultivar performance measure (*Pi*) was calculated as described by Lin and Binns [28] as:

$$Pi = \sum_{j=1}^{q} \frac{(Xij - Mj)2}{2q} \tag{7}$$

where *Pi* of a cultivar *i* under a CS or agro-environment *j* is defined as the mean square between the cultivar´s GY (*Xij*) and the maximum harvested GY (*Mj*) in the CS (*j*), averaged over the total number of CS (q) for the three years of trial. The smaller the mean square,

higher achievement in GY is the cultivar. The *Pi* values were ranked and twenty-two cultivars with smallest *Pi* values were selected in each of the three CS.

Thereafter, the GY stability under each CS was carried out with 46 consistently high yielding cultivars previously selected with *Pi* measurements. The stability of these cultivars was ascertained under each CS in the three years taken as environments. A combined analysis with the three CS over three years was carried out to estimate the cultivars´ stability performance under all CS. Cultivar stability index was estimated using the GEA-R software program as described by Pacheco et al. [29]. The Francis coefficient of variation CV (%) and the mean value were used as stability and performance indices, respectively [30]. With this approach, cultivars with high GY and low CV across environments were considered high yielding and stable.

The additive main effects and multiplicative interaction (AMMI) model analysis of variance for GY from the 46 selected high yielding cultivars was performed by GEA-R [29] to evaluate the cultivars and CS interactions. The model was

$$Yij = \mu + gi + ej + \sum_{i=1}^{N} \tau_n \gamma_{in} \delta_{jn} + \varepsilon_{ij} \tag{8}$$

where *Yij* is the yield of the $i^{th}$ cultivar ($i$ = 1, .., I) in the $j^{th}$ yearly CS considered as environment ($j$ = 1, ..., J), μ is the grand mean, *gi* and *ej* are the cultivar and CS deviations from the grand mean, respectively, $\tau_n$ is the eigenvalue of the PC analysis axis *n*, *γin* and *δjn* are the cultivar and environment principal components scores for axis *n*, *N* is the number of principal components retained in the model and *εij* is the error term. GGE biplots were generated by R with the Package *GGEBiplotGUI* using the first two principal components (IPCA1 and IPCA2) that explained the higher variation in the AMMI analysis for visual interpretation of G×E interaction.

The CS were classified based on the predicted means obtained from the AMMI analysis and the Biplot were visualized. Thereafter, the function "which-won-where" was used to identify best cultivars and group CS with high similarity. The biplots were based on an environment-centered (centering = 2) G by E table without any scaling (scaling = 0). It was environment-metric preserving (SVP = 2) and the axes were drawn to scale (default feature of GGEbiplot).

### 2.8. Cropping System Performance and Discriminating Level

We classified CSs based on their performance indices and identified the least discriminative CS. The least discriminative CS is the CS where cultivars better utilized the available nitrogen. We calculated two CS performance indices. The first used the whole set of 220 cultivars, while the second used the set of 46 high yielding cultivars. The CS performance indices were calculated using the following formula.

$$P(cs) = \sum_{j=1}^{n} \frac{Pi}{n} \tag{9}$$

where *P(cs)* is the performance index of the CS, *n* is the total number of considered cultivars and *Pi* is the performance index of *jth* cultivar as described by Lin and Binns [28].

## 3. Results

### 3.1. Nitrogen and Fungicide Application Effect on Phenotypic Traits

The phenotypic traits expression differed significantly ($p < 0.001$) among genotypes and CS (Table S5). The interactions of G×CS across the three growing seasons were also significant ($p < 0.001$). Means comparison by Tukey HSD test showed that the cultivars grown under HN-WF and HN-NF showed better performance for most traits evaluated than cultivars grown under LN-NF (Figure 1; Figure S2; Table S6). GY ranged from 6.455 Mg·ha$^{-1}$ under LN-NF in 2016 to 11.225 Mg·ha$^{-1}$ under HN-WF in 2015 (Table S7).

GY was increased due to N, NF and F applications, with a range from 6.39% for $N_{eff}$ in 2016 to 68.03% for $NF_{eff}$ in 2015 (Figure 2A,C; Tables S6 and S7). Results indicated that kernel number per $m^2$ (KNms) was highly increased by N (30.24% in 2016 and 27.94% in 2017) and by NF (53.21% in 2016 and 32.81% in 2017) application, while TKW was reduced by N and NF applications (Table S6).

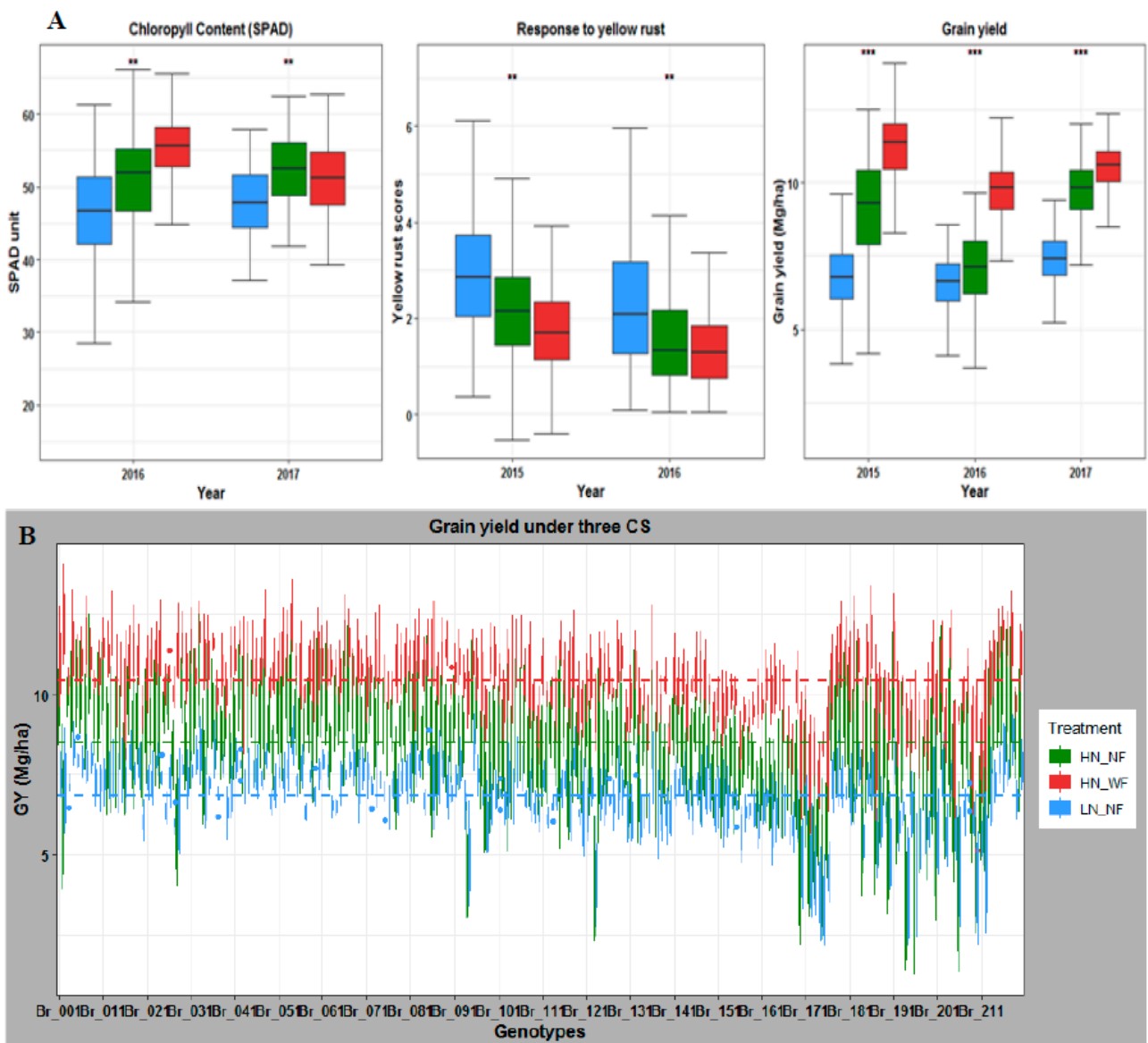

**Figure 1.** Differences between the three cropping systems. (**A**) Leaf chlorophyll content, plant response to yellow rust, and GY traits. (**B**) Three years averaged grain yield production (Mg·ha$^{-1}$) of 220 cultivars under the three cropping systems. The dashed lines indicate the GY mean under each CS: blue: LN-NF (6.850 Mg·ha$^{-1}$); green: HN-NF (8.507 Mg·ha$^{-1}$), and red: HN-WF (10.447 Mg·ha$^{-1}$). The symbol *, **, and *** meansignificant at $p$ = 0.05, 0.01, and 0.001, respectively.

SPAD values increased under HN-NF and HN-WF in 2016, indicating an increasing effect of nitrogen and fungicides on leaf chlorophyll content. However, a decrease (−2.2%) of chlorophyll content due to fungicide effect was observed in 2017. The yellow rust (YR) effects on the cultivars were reduced by 26.59% and 32.3% under HN-NF compared to LN-NF in 2015 and 2016, respectively. In addition, the cultivation under HN-WF reduced plant rust infection by 41.29% in 2015 and 38.08% in 2016. In 2017 season, the infection of yellow rust was not observed; therefore, it was not scored. Regarding grain quality traits, N, NF, and F had an increasing effect on grain crude protein content with NF having the

highest effect followed by N effect. Sedimentation volume was significantly increased by N application over the three growing seasons, whereas it was not affected by fungicide application in 2015 and 2017. However, grain starch content was decreased by nitrogen and fungicide applications (Table S6).

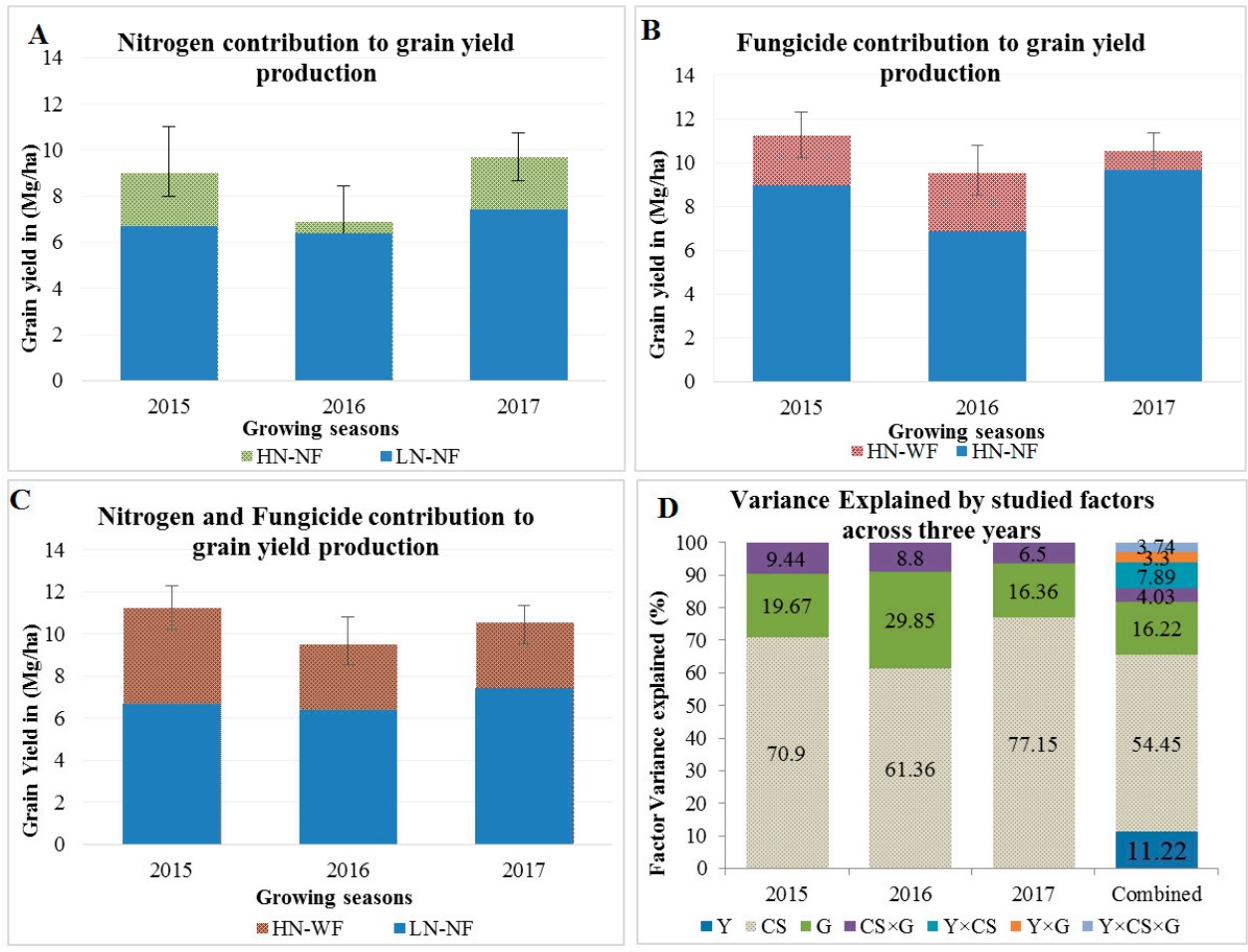

**Figure 2.** Contribution of nitrogen and fungicide to grain yield (Mg·ha$^{-1}$) production in 2015, 2016, and 2017 growing seasons and variance components of each factor. (**A**) Blue bars show the grain yield harvested under LN-NF and green/white bar above show grain yield gained from additional fertilization. (**B**) Blue bars are grain yield from HN-NF used as control and the above red/white show gained grain yield owing to fungicide application. (**C**) Blue bars are grain yield harvested under LN-NF (Control) and the above red/green bars show gained grain yield due to combined effect of additional nitrogen fertilization and fungicide application. (**D**) Proportion of the variance components of factors years (Y), cropping systems (CS), and genotypes (G) and their interactions.

Coefficient of variation of evaluated traits ranged from 0.237% for kernels per spike in 2017 to 62.73% for YR in 2016. Heritability estimates (H$^2$) ranged from 0.22 for biomass in 2015 to 0.95 for heading date (HD) in 2017. H$^2$ estimates for YR were consistently high with 0.84 in 2015 and 0.92 in 2016 (Table S5).

We further investigated yield performance through three way ANOVA and variance components analysis with year (Y), Genotypes (G) and CS as factors, to evaluate the effect of each factor and identify the highest source of variation in yield. Results revealed significant ($p < 0.001$) differences among years, cropping systems, genotypes and their interactions (Tables S8 and S9). The cropping system was the most important source of variation in grain yield, with 54.45% followed by the genotypes and years, explaining 16.22% and 10.38% of the total variance, respectively (Figure 2D; Table S9).

### 3.2. Application of Nitrogen and Fungicide Improve GY and Grain N Yield (GNY)

The NUE and NAE estimates were used to examine the proportion of N used for grain production. The results indicated that both estimates varied significantly ($p < 0.001$) among genotypes and CS in the three growing seasons. Significant G × CS × Y interactions were also observed (Table S10A). NUE was twice that under LN-NF than under HN-NF and HN-WF, indicating that higher amounts of N fertilization did not lead to an increase in NUE. The estimates of NAE were significantly higher under HN-WF when compared to the HN-NF (Figure 3). Moreover, the classification of cultivars based on their NAE values revealed that a total of 14, 10 and 73 cultivars were efficient N utilizers nitrogen for grain production under HN-WF in 2015, 2016, and 2017, respectively. However, under HN-NF, only 12 cultivars efficiently utilized the available nitrogen in 2017 (Figure S3). The grain nitrogen yield (GNY) differed significantly among cultivars, CS and years. Similarly, a significant Y × G × CS and G × CS interaction was detected for GNY (Table S10A). Compared to LN-NF, GNY increased under HN-NF and HN-WF in all growing seasons. Nitrogen plus fungicide increased the GNY compared to the system when N was supplied alone across the three years of trials (Figure S4A). GNY strongly correlated with GY under all CS (HN-NF = 0.98; HN-WF = 0.90, LN-NF = 0.97) (Table S10).

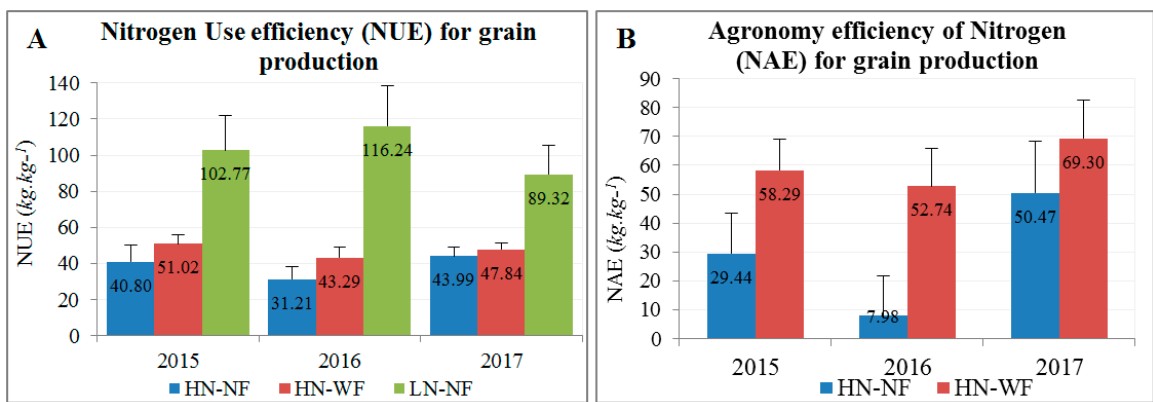

**Figure 3.** (**A**) Nitrogen use efficiency. (**B**) Agronomy efficiency of nitrogen in three years of trial.

### 3.3. Genetic Relationship between GY and Evaluated Traits

The genetic relationship between GY and each traits of interest across cropping systems was evaluated through Pearson correlations coefficients and linear regression analysis based on cultivar means. For cultivars response to yellow rust infection, we noticed that an increase in the disease infection reduced the GY in 2015 (r = −0.52***) and 2016 (r = −0.51***) seasons (Figure S5). The response to YR explained 36.4%, 31.7%, and 11.14% of the variation in GY under LN-NF, HN-NF and HN-WF, respectively. Under LN-NF and HN-WF, GY decreased 0.5333 and 0.4517 Mg·ha$^{-1}$ per unit increase in YR infestation, whereas, GY reduction was significantly higher under HN-NF, with 0.9665 Mg·ha$^{-1}$ decrease per unit increase in YR infestation compared to HN-WF and LN-NF (Figure 4A; Table S11). Cultivars with low YR infestation recorded higher yield under all three CS (Figure S4B).

The correlation and regression analyses performed showed significant and positive relationships between GY and leaf chlorophyll content (SPAD) in 2016 (r = 0.62***) and 2017 (r = 0.33***) (Figures S5 and S6A,B). Independent of the CS, leaf chlorophyll content explained 41.52% and 3.56% of the variation in GY in 2016 and 2017, respectively. The increase in GY was estimated to 0.2098 Mg·ha$^{-1}$ in 2016 and 0.0747 Mg·ha$^{-1}$ in 2017 per unit increase in SPAD value (Figure S6A,B). Leaf chlorophyll content more explained the variation in GY under LN-NF ($R^2$ = 0.4196 in 2016 and $R^2$ = 0.059 in 2017) than under HN-NF and HN-WF (Figure 4B,C). In 2016, GY increased equivalently under LN-NF and HN-NF with 0.1332 and 0.1518 Mg·ha$^{-1}$ per unit increase in SPAD, respectively, and both were significantly higher than the one observed under HN-WF, which amounted

to 0.0717 Mg·ha$^{-1}$·SPAD$^{-1}$ in the same year (Table S12). Cultivars with higher SPAD obtained higher GY than those with lower SPAD values under LN-NF (Figure S4C).

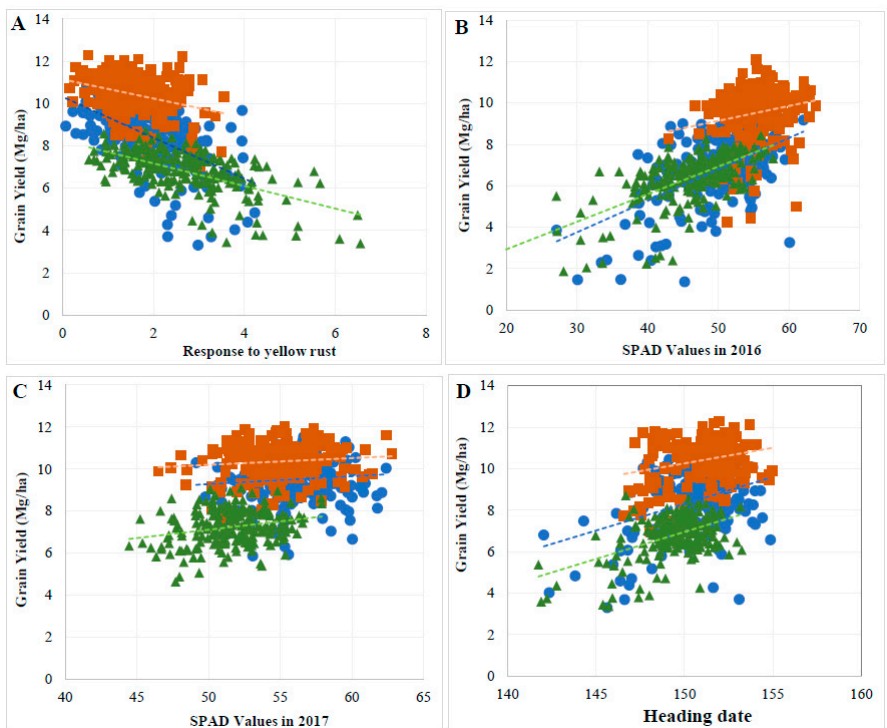

**Figure 4.** Relationship between GY and traits of interest under three CS. (**A**) Response to yellow rust scored in 2015 and 2016 seasons. (**B**) SPAD in 2016. (**C**) SPAD in 2017. (**D**) Heading date of three years averaged per cultivars. Color-shape symbols: green-triangle for the cropping system LN-NF, blue-circle for HN-NF, and red-square for HN-WF. The regressions equations, the significance of the slopes and comparison among slopes of the three CS is given Table S12.

HD was positively correlated with GY (r = 0.19**, r = 0.33***; r = 0.41*** in 2015, 2016 and 2017, respectively), while negative correlations were detected between GY and PH (r = −0.46***, r = −0.26*** and r = −0.51*** in 2015, 2016 and 2017, respectively) (Figure S5). A longer vegetative period due to delay of HD was beneficial in increasing cultivar GY, most importantly under LN-NF with an increase of 0.2607 Mg·ha$^{-1}$ per one additional day. The delay in HD equally affected GY under HN-WF and HN-NF (Figure 4D; Table S12). On the other hand, the increase in PH reduced GY under all three CS, including a pronounced yield reduction under HN-WF with 0.0669 Mg·ha$^{-1}$ decrease per cm increase in PH. The reduction in GY per increase in PH was significant only when compared HN-WF to LN-NF (Figure S6C, Table S12).

GY positively correlated with most of the yield components across the three growing seasons (Figure S5). Result indicated that GY was positively and significantly ($p < 0.001$) correlated with kernels per spike (r = 0.35–0.42), and KNms (r = 0.52–0.53). GY exhibited positive correlation with HI (r = 0.59–0.79) and with PBWms (r = 0.41–0.64) across the growing seasons. Under each CS, GY recorded significant and positive correlations with its key components, especially with KNms (r = 0.36–0.56), except for SNms and TKW (under HN-WF) as shown in Figure 5. Like KNms, GY harvested under one CS positively and significantly correlated with those from the two other CSs (Figure S5D,E). Results of the comparisons among correlations (GY vs its components) coefficients from the three CS indicated significant ($p < 0.001$) differences. Correlation coefficients obtained under HN-WF were different with HN-NF and LN-NF for PBWms, TKW and KNms. The cropping system did not affect the relationship between SNms and GY as shown by insignificant differences among correlations coefficients (Table S11).

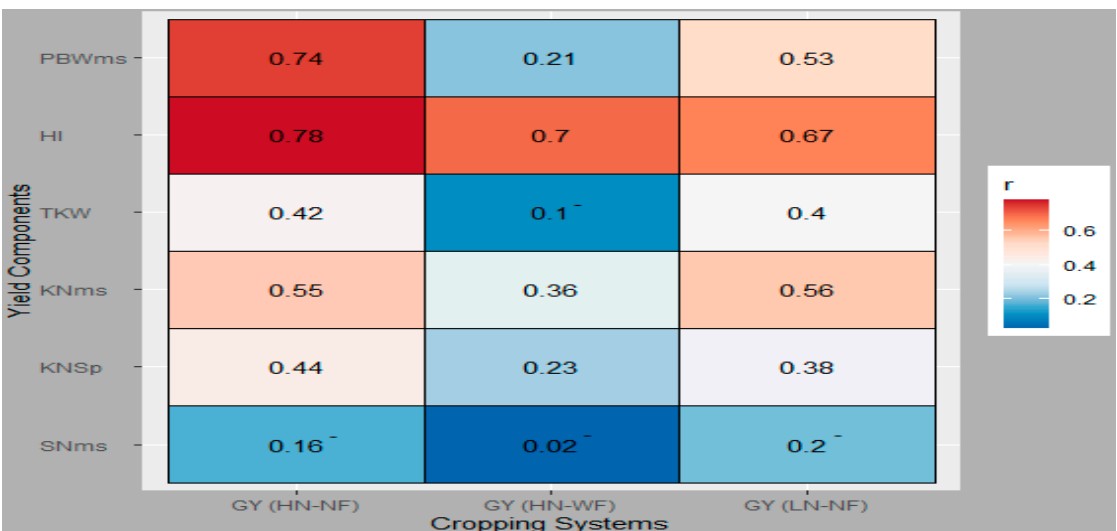

**Figure 5.** Correlations between GY and yield components traits under each CS. Traits were measured in a wheat population containing 220 cultivars grown under three CS between 2015 and 2017. The correlations coefficients are ranged low values (blues color) to high values (red colored). The numbers crossed with minus $^{(-)}$ are not significant at $p < 0.05$.

The genetic link between GY and grain quality traits such as grain protein (GPC), grain starch (GSC), and sedimentation investigated revealed significant positive associations between GY and GSC ($r = 0.34–0.65$, $p < 0.001$) across growing seasons. However, GY negatively correlated with GPC ($r = -0.18^{**}$, $r = -0.46^{***}$, $r = -0.27^{**}$) and sedimentation ($r = -0.19^{**}$, $r = -0.22^{**}$, $r = -0.30^{***}$) across three years (Figure S5). The regression analysis revealed that the variation in GSC significantly explained the variation in GY across all three CS, but mostly under HN-WF with 48.13% of variance ($R^2$) explained in GY (Figure S6D). A unit increase in GSC enhanced GY by 0.8573, 0.7954 and 0.7765 Mg·ha$^{-1}$ under HN-NF, HN-WF, and LN-NF, respectively, whereas GY decreased by 1.5502, 1.0854 and 1.2532 Mg·ha$^{-1}$ when GPC increased by one unit under LN-NF, HN-NF and HN-WF, respectively (Figure S6E, Table S12). GY reduction per unit increase in GPC was significantly higher under LN-NF than under HN-NF and HN-WF. Therefore, the highest trade-off relationship between GY and grain protein content occurred under LN-NF.

*3.4. GY Was Significantly Affected by Indirect Effects of Several Agronomic Traits*

Full regression and path analysis were used to quantify the effect of several agronomic traits on GY (Table S13A). The full regression model captured 86.2, 81.6, and 84.7% of the total variation in GY under HN-NF, HN-WF and LN-NF, respectively. HD, SPAD, YR, HI, PBWms and GSC had significant ($p < 0.001$) effects on GY. As revealed by the path coefficients (Table S13B), most traits had higher indirect effects on GY than direct effects, except HI and PBWms, under all CS. The path correlation coefficients relating YR to GY were made of sizeable negative indirect effects (−0.405 under HN-NF; −0.179 under HN-WF and −0.359 under LN-NF) via other traits and direct effects (−0.158 under HN-NF; −0.128 under HN-WF and −0.243 under LN-NF).

*3.5. Few Cultivars Achieved Maximum Yield under HN-NF, the Most Discriminating CS*

We defined the least discriminative CS as the CS under which most cultivars had their GY close to the highest yielding cultivar, hence having a CS performance index (P*cs*) close to zero. P*cs* describes how well cultivars come to achieve the maximum yield potential under a CS. The Pcs calculated with 220 cultivars (Figure 6A), and with 46 high yielding cultivars (Figure 6B), showed similar trends in the performance of the CS. Averaged over three years, LN-NF recorded the lowest performance index and, therefore was more suitable for many cultivars to reach a GY close to the maximum yield. HN-WF was the second least discriminative CS, while HN-NF recorded the highest Pcs indices calculated with 220 cultivars

(12,154) and with 46 cultivars (591), and was the most discriminating CS. Further, biplots from AMMI analysis revealed the discriminating ability and the representativeness of tested CS taken as different environments (Figure 6C). The two first principal components of the biplots explained 83.67% of the total variation of the environment-centered G by E table (G + GE) and revealed that LN-NF was more representative of other tested CS, and thus less discriminating. The length of the vectors of HN-NF was greater than LN-NF and HN-WF, indicating that HN-NF had the highest discriminating ability. Thereafter, the function "which-won-where" identified high similarity between LN-NF and HN-NF, making one mega environment with cultivars Hybery and Tabasco as winning cultivars. The cropping system HN-WF was considered as one mega environment with Tobak as winning cultivars.

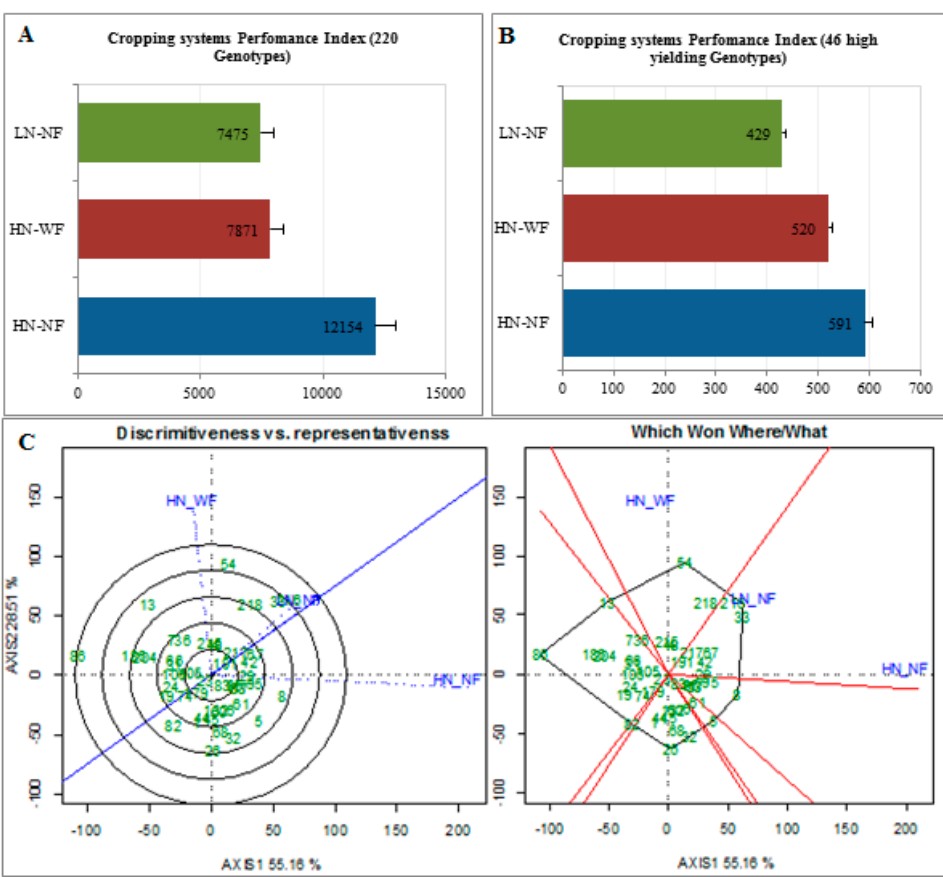

**Figure 6.** Graphical representation of CS performance index of (**A**) 220 cultivars and (**B**) high yielding cultivar. (**C**) Biplot of discriminativeness vs. representativeness of test environments (left panel) and the identification of winning cultivars and their related CS (right panel).

### 3.6. High Yielding Cultivars Showed Different Stability Level under the Three Cropping Systems

The performance index of Lin and Binns (1988) was used to identify high performing cultivars across growing seasons under different CS. A total of the highest yielding 81 cultivars were selected across the three CS over the three growing seasons. Among them, 46 were selected at least under five agroenvironments, and comprised 43 newer and three older released cultivars (Figure S8). Results of GY stability (Francis CV) of these 46 high yielding cultivars are presented in Figure S9 and Table S14. Among them, 19 recently released (after 2000) cultivars showed higher GY and higher stability index (low CV) under HN-NF, while 14 (all recently released) and 10 (with eight newer and two older) were stable under HN-WF and LN-NF, respectively. Fourteen cultivars (13 newer and one older) exhibited high stability when considering combined three CS (Table S14). GY average values of stable cultivars ranged from 7.866 Mg·ha$^{-1}$ for cultivar Hyland under LN-NF

to 12.288 Mg·ha$^{-1}$ for Tobak under HN-WF, while the lowest and the highest CV were obtained by Edward under HN-WF (0.2543%) and Atomic under combined CS (18.80%). A Venn diagram revealed that only the cultivar Hyland out of the 32 stable cultivars was high performing and stable under all CS. Among the other 31 cultivars, seven were stable under two CS (Table S14, Figure S9).

### 3.7. Newer Released Cultivars Outperformed Older Cultivars in Most Traits and GY under All CS

The breeding progress in the studied wheat panel was quantified by the slope of the regression between years of release of 209 cultivars and the values of traits of interest. Under each CS, breeding progress for all traits evaluated were significant, except for HD and SNms. Breeding increased GY via increasing the key yield components, most importantly on KNSp and KNms, while reducing the PH and the response to disease over years (Figure 7; Figure S7). For SPAD and YR, significance differences of the breeding progresses among the three CS were detected, which indicate that cultivars behaved differently under the three CS depending on their years of release. The highest increase in breeding progress trends for SPAD was observed under LN-NF, while the breeding progress for YR observed under HN-WF was the weakest among the three CS (Table S15). For GY, the breeding progress was significantly higher under HN-NF than both HN-WF and LN-NF, where the breeding progresses recorded similar trends (Figure 7A). Breeding also decreased the coefficient of variation of GY, and increased the yield stability (Figure 7B). KNms and sedimentation showed slow breeding progress under LN-NF than under HN-NF and HN-WF (Figure S7). The correlations between cultivars years of release and the evaluated traits were significant and almost equal under all three CS, suggesting that breeding had increased the cultivar adaptability under low, semi, and high input CS (Figure 7C).

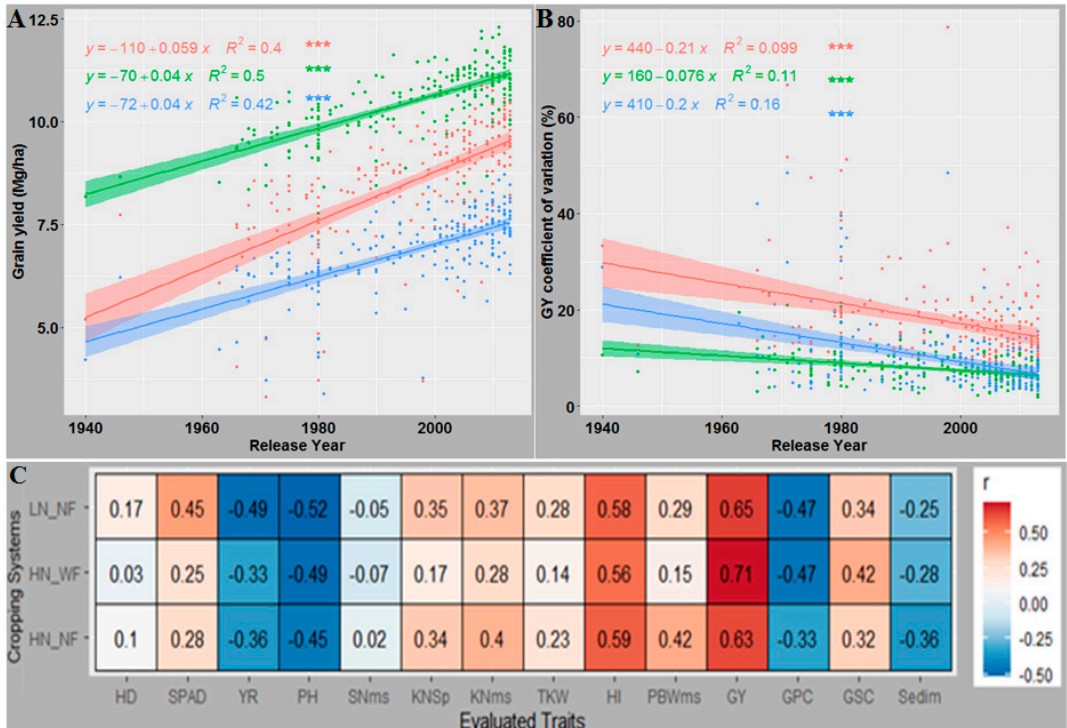

**Figure 7.** Temporal trends observed in traits of interest. (**A**) Average GY of three years per cultivars. (**B**) Average GY coefficient of variation per cultivars in relation their years of registration under three CS, HN-WF in green, HN-NF in red and LN-NF in blue color. (**C**) Correlations coefficients between year of release and fourteen traits of interest under three CS. The center lines represent the regression lines and the shaded regions in (**A**,**B**) represent the 95% confidence intervals. *** means the *p* value is significant at 0.001.

## 4. Discussion

### 4.1. Variation in Agronomic Traits and GY under the Effect of Nitrogen and Fungicide

Nitrogen and fungicide application increased wheat yield. Higher GY following N fertilization was due to the increase of grain number [31,32]. GY improvement under intensive CS could be explained by a synergistic effect of nitrogen and fungicide to increase grain numbers (kernel per m2, kernel per spike). Fungicide supply positively contributed to GY owing to the maintenance of green leaf area, particularly flag leaf life extension after anthesis, affecting photosynthate partitioning within the plant, which enhances grain weight [33,34]. In fact, fungicide application contributed to plant protection against disease invasion, particularly YR, which was more pronounced in the two first years of trial and very known for its effect on yield loss [35]. In 2017 growing season, the infection of yellow rust was not observed; hence it was not scored. The lower infestation of yellow rust could be related to the weather conditions at the experimental site (Figure S1), which were drier in 2017 with higher temperatures and lower rainfall compared to the 2016 and 2015 growing seasons. Lower temperatures and high relative humidity increase the infection levels of yellow rust [36]. Nitrogen application enhanced plant resilience to YR infestation, whereas earlier studies reported higher rust infestation under high N application rates owing to increased plant canopy size [12,13,37]. Our result may be due to the fact that we supplied N fertilization in the form of ammonium, which decreases stem and YR, while nitrate-N increases them [38]. Plant height was decreased under nitrogen and fungicide supply due to the application of growth regulator hormones in these CS [39].

Most evaluated traits recorded moderate to high heritability ($H^2$) estimates across the three growing seasons, and thus can be exploited to identify QTLs underlying variation in traits in the evaluated diversity panel using the GWAS approach [1]. GY recorded high $H^2$ across all growing seasons, suggesting that selecting for high yielding cultivars for each CS could be simplified in our panel; particularly TKW with $H^2 = 0.84$ were observed under several nitrogen supply experiments [40].

### 4.2. Nitrogen and Fungicide Application Effects on N Flow from Soil to the Grain

Nitrogen and fungicide led to increased leaf chlorophyll content and grain quality. Under high N input CS, the amount of nitrogen molecules required for the constitution of chloroplasts is satisfied, and sufficient nitrogen would result in high chlorophyll content [41]. Besides, nitrogen has a large effect on leaf growth, increases leaf area, and the intensity of photosynthesis [42]. In line with the present study, Bryson et al. [13] reported that fungicide sprayed on wheat plants increased chlorophyll content of leaves.

Nitrogen and fungicide are contributing to high GPC and GNY, owing to high N relocation from the shoot to the growing grain [33]. Fungicide application increased grain N accumulation through the improvement of nitrogen uptake from soil and the remobilization of nitrogen from plant green tissues to the grain [43]. We obtained 13.95%, 13.66%, and 11.17% GPC under HN-WF, HN-NF and LN-NF. However, the standard minimum GPC for bread baking is 12%, which requires an amount of 180 kg·ha$^{-1}$ fertilizer N [31]. The lowest grain crude protein (GPC) and grain N yield (GNY) was recorded under LN-NF CS, which would indicate a deficiency of most cultivars to remobilizing N to the grain in the absence of sufficient nitrogen and fungicide (Table S15a). Even under sufficient N, the use of a high yielding cultivar is a requisite to obtain higher GNY (Table S15b). Hawkesford [8] reported that only a third of nitrogen inputs to cereal crops worldwide are recovered in grain for consumption. These findings attest the necessity to breed for cultivars with improved N use efficiency and confirm that high nitrogen supply is not synonymous with high yields. Cultivation of cultivars has adapted and selected for low input CS i.e., organic farming has resulted in improvement of GY under this CS [44]. However, growing of less N uptake efficient cultivars will result in a high amount of residual N. Likewise, higher GNY obtained under HN-WF compared to HN-NF, although they had the same amount of N, indicated the existence of important residual N that was not removed from HN-NF.

Ladha [45] reported that wheat harvested total N comprised 48% of applied fertilizer N and 52% of other sources like nonsymbiotic N2 fixation (24%), manure (14%) and atmospheric deposition (6%). Therefore, almost half of the 155, 165 and 135 kg·ha$^{-1}$ N fertilizer applied in 2015, 2016 and 2017, respectively (Table S2) was recovered in grain, resulting in a loss of 80.6, 85.8 and 70.2 kg·ha$^{-1}$ N fertilizer in the agro ecosystem through leaching, runoff or erosion [44]. Nitrogen loss from N fertilizer is increasing with the increasing use of applied fertilizers [44]. Angus [46] reported ~20 kg·ha$^{-1}$ N (including all sources of N) is needed to produce one tone of wheat grain, and 6 kg·ha$^{-1}$ N for one tone of straw production. According to this benchmark of 20 kg·ha$^{-1}$ N, under our field conditions a total of 25.86 and 21.07 kg·ha$^{-1}$ N was utilized to produce one tone of grain out of the available 220 kg·ha$^{-1}$ N for HN-NF and HN-WF, respectively, whereas 9.56 kg·ha$^{-1}$ N was utilized out of 65.43 kg·ha$^{-1}$ for LN-NF (Table S15c). Therefore, a greater amount of N was required under HN-NF to produce the same quantity of grain compared to HN-WF and LN-NF, showing low NUE under HN-NF. Moreover, nitrogen loss in the agro-ecosystem following over-fertilization, which decreased the NUE, has been reported [8]. In a previous report on global nitrogen budgets in cereals, the total amount of N input in the agro-ecosystem consisted of 51% of N fertilizer, and 15% and 19% of biological fixation and manure, respectively, followed by crop residue (8%) and deposition (about 7%). Considering the importance of N in increasing GY, and the environmental damage following huge rates of N application, combining N application with fungicide [37], using the optimum amount of N [2], or exploring other sources of N like biological N$_2$ fixation, has lately raised interest in agricultural research [47]. The estimates of NAE was significantly higher under HN-WF when compared to HN-NF, implying that fungicide had an increasing effect on NAE, demonstrating a synergic effect of N and F on cultivars performance to produce grain.

A trade-off relationship was observed between GY and grain protein content (GPC). The negative correlation between GY and grain quality parameters, particularly GPC is well known and constitutes a constraint in breeding for high GPC and GY [48]. Genetic factors may be responsible for the undesirable associations between these traits [49,50]. This could be a consequence of a dilution effect causing a reduction in GPC as yield increases [40]. To select high GY while maintaining high GPC, the deviation from the GPC–yield relationship (GPD) has been suggested as a metric for selection [51]. The GPD is related to postanthesis N uptake, and might be associated with genotypic differences in access to soil N [52]. Under low N CS, nitrogen would primarily be allocated to GY, which was a primary breeding goal in cereals during last decades [53]. However, eleven cultivars comprising three new released and six old cultivars, and two unknown released years were among the low yielding cultivars and had GPC of 12.14%, 14.40 and 15.03% under LN-NF, HN-NF, and HN-WF, respectively.

### 4.3. Agronomic Traits Contribution to GY across Cropping Systems

Most of the agronomic traits showed higher indirect effects than direct effects on GY, except HI under the three CS, and biomass under HN-NF and HN-WF, in which direct effects were significantly higher than previously reported [54]. Our results revealed that GY was more influenced by kernel number per m$^2$ rather than kernel weight and number of spikes among its components, independently of the CS. Nevertheless, Mansouri et al. [54] reported spike number as the most significant variable influencing GY under south Mediterranean conditions. However, low correlation observed between spike number and GY in some cases is due to the number of infertile florets on spikes [34]. The delay in HD was beneficial to high GY under the three CS, importantly under LN-NF. Early HD of a cultivar may lead to early maturity, which is a limiting factor for high GY due to the reduced time period for assimilate translocation to the grain. Increased GY obtained from late-heading cultivars is attributed to the increased number of fertile florets as a result of higher assimilate accumulation during the preflowering period [34].

SPAD index measured at the heading growth stage as an indication of leaf nitrogen content was positively correlated with GY, as reported in several studies [55,56]. The variation in the leaf chlorophyll content explained the variation in GY under LN-NF, and cultivars with higher SPAD obtained higher GY under this CS. These results showed that leaf chlorophyll content under low nitrogen is a physiological indicator to select nitrogen uptake and utilization efficient cultivars.

GY was negatively correlated with the response of cultivars to YR infestation, consistently across CS. Similar to our results, it has been reported that YR disease is a major cause of wheat yield loss worldwide [57]. Although YR infestation under HN-NF was lower compared to LN-NF, its effect on GY reduction under the former was greater. Similarly, previous research reported significant negative effects of YR on GY under high nitrogen input CS. High nitrogen increases crop size and canopy density, which creates favorable conditions for YR invasion [12,13]. To control the negative effect of YR and other diseases [43] the use of fungicide and YR-tolerant cultivars is a valuable resource to increase GY.

### 4.4. Breeding Progress in Agronomic Traits and Yield Stability across Cropping Systems

The regression results, together with the correlation between traits and cultivar release year, provide evidence that breeding has enhanced cultivar performance not only under optimal conditions but under production systems with reduced agrochemical inputs. Breeding has accumulated genetic variants conferring favorable effects on key yield parameters, photosynthetic activity and disease resistance, which subsequently have enhanced GY [16]. GY was negatively correlated with PH, which can be due to the reduction in PH through incorporating the dwarfing genes (*Rht1 (Rht-B1b), Rht2 (Rht-D1b), Rht-D1c, and Rht8*) in modern high-yielding cultivars [58].

The contribution of breeding to GY was not related to the increase in HD, because as opposed to GY, day to heading has increased in the breeding history, even though, a strong correlation occurred between vegetation duration with high GY. Although not shown, our results revealed a low decline in the shoot dry weight over years. Similarly, it has been reported that breeding improved GY by increasing the HI through allocation of resource to grain number per m$^2$ rather than shoot biomass production [59]. Despite being significant, the breeding progress in TKW has been lower compared to other yield components. Similar results were found by Lichthardt [27], reporting that TKW was not affected by breeding, unlike other traits such as green canopy duration and other source components that have increasing effects on GY.

The highest yielding cultivars across CS comprised only three old and 43 recently released cultivars registered in/or after 2000. This result highlighted the tremendous role played by breeding in increasing GY. The improved cultivars showed differences in their stability levels under the three CS, indicating high G×CS interaction, and support the necessity to use convenient cultivars for targeted CS. Kadhem and Baktash [60] reported that selecting for promising cultivars must include the criteria of high yield and the stability performance of the cultivar, because some cultivars may be high yielding but unstable across growing seasons and/or CS. We defined the least discriminative CS as the one with most cultivars having their GY means close the highest yielding cultivar, i.e., a CS with a performance index (P$cs$) close to zero. HN-NF was the most discriminative CS, indicating that most cultivars could not make use of the available nitrogen fertilizer. Under HN-NF CS, N use efficient cultivars are more likely to achieve high yield. It has been recommended to use superior cultivars for the best use of the available nitrogen and other resources to avoid resource waste [61].

The results of AMMI analysis are very useful in determining specific adaptation, cultivar stability and choice of the best CS [62,63]. The association of LN-NF and HN-NF into one mega CS could be explained by the absence of fungicide, which was a determinant factor in the agronomy efficiency use of nitrogen. Under these two CS, many cultivars could not adequately use the available nitrogen and express their yielding potential. Further studies should be done with several nitrogen application levels to identify the optimum

level of N input to obtain better GY and grain quality, as 65.43 kg·ha$^{-1}$ was too low, while 220 kg·ha$^{-1}$ was too high.

## 5. Conclusions

The present study revealed that leaf chlorophyll content measured by a SPAD meter around heading growth stage could serve as a proxy to estimate GY under low nitrogen conditions. The results showed that nitrogen and fungicides have synergistic effects on GY production and grain protein content (GPC). In the absence of fungicide application, YR greatly decreased GY, mostly under high nitrogen input CS. Under our field conditions, most cultivars obtained an average GY close to the cultivars with maximal yield under LN-NF and HN-WF compared to HN-NF. HN-NF was more selective in GY production, as few cultivars were close to the highest yielding cultivar. HN-WF achieved the best GY production because of fungicide, which played an important role in extending plant life cycle and photosynthesis activity. Among 46 high performing cultivars used in the stability analysis, 19, 14, and 10 cultivars were stable under HN-NF, HN-WF and LN-NF, respectively. The leaf chlorophyll content and the cultivar resilience to YR infection played an important in enhancing GY. Therefore, selection for these traits and identification of genetic factors underlying them could be considered in wheat breeding programs and in future genetic studies to improve GY. AMMI Analysis confirmed the discriminating power of HN-NF, indicating that fewer cultivars could make use of the additional fertilizer. These results suggest and recommend the cultivation of nitrogen use-efficient cultivars, and to associate different nitrogen levels with fungicide to maximize nitrogen use and avoid resource waste. New breeding strategies for high GY should promote selection of cultivars for specific CS, i.e., for high and low input CS, and include leaf chlorophyll content and resilience to YR as selection criteria. Released cultivars should be labeled with their NUE level and its favorable CS to enable organic or conventional agriculture farmers to make a better choice when growing a cultivar.

**Supplementary Materials:** The following are available online at https://www.mdpi.com/article/10.3390/agronomy11071295/s1, Figure S1: Weather conditions, rainfall and temperature data from the experimental site, Figure S2: Significant differences among the three cropping systems (CS) for evaluated traits with LN-NF in gray, HN-NF in green, and HN-WF in red color, Figure S3: Agronomy efficiency use of nitrogen supplied, Figure S4: (A) Grain harvested N yield (GNY) under three CS (HN-NF in green color, HN-WF in red color, LN-NF in blue color) across the three years; (B) Resilience to YR infestation of GY contrasting cultivars; (C) Chlorophyll content (SPAD) of GY contrasting cultivars, Figure S5: Pearson correlation coefficients and associated probability among evaluated traits based on of the cultivar mean from the three tested cropping systems in three years of trials, Figure S6: Relationship between GY and traits of interest, Figure S7: Temporal trends observed in evaluated traits in relation to year of registration among 209 cultivars under three CS, HN-WF in green, HN-NF in red and LN- NF in blue color, Figure S8: Graphical representation of selected cultivars based on performance index, Figure S9: Biplot of the coefficient of variation (Y-axis) plotted against mean yield (X-Axis) of GY of 46 winter wheat cultivars under HN-NF, Table S1: Year-wise soil information of the experimental site, Table S2: Fertilizer application, amount and the developmental stage of crop, Table S3: Application of fungicides/pesticides on different developmental stages of the wheat, Table S4: Description of the measured variable in the experiments, Table S5: Summary of analysis of variance of agronomic and grain quality traits of 220 cultivars tested in three different environments, Table S6: Arithmetic mean and treatments effect of agronomic and grain quality traits of 220 cultivars tested in three different CS over three growing seasons, Table S7: GY (Mg·ha$^{-1}$) statistics for the applied cropping systems and the years of experiments, Table S8: Detailed analysis of variance of GY of winter wheat genotypes in cropping systems (CS) by year (Y), Table S9: Three way ANOVA of GY of winter wheat genotypes (G) in three cropping systems (CS) across three years (Y), Table S10: N flow related analysis of variance and statistics, Table S11: Pairwise comparison of GYs and its components correlation coefficients among cropping systems, Table S12: Pairwise comparison of coefficients (intercepts and slopes) of regressions model GY vs traits of interest under three CS, Table S13: Full regression (A) and path models (B) with direct and indirect effects of 13 independent variables on GY of 220 cultivars tested in three different CS over three growing seasons, Table S14:

Summary of winning genotypes in the three environments and their year of release, Table S15: Pairwise comparison of coefficients (intercepts and slopes) of regressions model traits of interest vs. years of release under three CS.

**Author Contributions:** Conceptualization, A.P.K., M.M.B. and A.B.; data curation, A.P.K.; formal analysis, A.P.K.; funding acquisition, J.L. and A.B.; A.P.K. and M.M.B.; methodology, A.P.K., M.M.B., B.C.O. and A.B.; project administration, J.L. and A.B.; resources, J.L. and A.B.; Software, A.P.K. and J.L.; supervision, J.L. and A.B.; validation, J.L. and A.B.; writing—original draft, A.P.K. and M.M.B.; writing—review & editing, A.P.K., M.M.B., B.C.O., J.L. and A.B. All authors have read and agreed to the published version of the manuscript.

**Funding:** This research was funded by the German Federal Ministry for Research and Education (BMBF), IPAS Program with grant number 031A354C.

**Data Availability Statement:** All the produced data are available and are submitted with the manuscript including the supplementary files.

**Acknowledgments:** We thank the PROJECT MANAGEMENT JÜLICH (PtJ) for their support. The doctoral study of P. Koua was funded by the Konrad Adenauer Foundation. Special thanks to the team of the Campus Klein-Altendorf for the maintenance of the experiment.

**Conflicts of Interest:** The authors declare that the study was conducted in the absence of any commercial or financial relationships that could be envisaged and/or construed as a conflict of interest.

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
