# Peer review of "Fungicide Application Affects Nitrogen Utilization Efficiency, Grain Yield, and Quality of Winter Wheat"

_agronomy, doi:10.3390/agronomy11071295_

Round 1

Reviewer 1 Report

The manuscript authored by Koua et al. assessed the interaction effect of N and fungicide application on grain yield and quality of wheat through a 3-year field experiment based on 220 cultivars. Here are my specific comments:

  1. L18: full name of CS should be given in the abstract
  2. L73: Figure 1 should be given in the section 2.1
  3. Table S2: full name of the abbreviations “KAS”,”BBCH”should be given. Nutrient component of the fertilizer should be given
  4. I suggest change the unit of GY from dt.ha-1 to kg ha-1 or Mg ha-1
  5. Please add unit for each trait in the Table S6
  6. Font in Figure 8 should be larger
  7. I do not think L408-412 is necessary
  8. L550-551: This is not a conclusion
  9. Figure 5 is lack of self-explanatoriness. Why there was no relationship between GY and SPAD in the growth season 2014-2015? Figure 5 a and d were based the data of which year?
  10. Why the data of the growth season 2014-2015 is missing from the first figure of the Figure 2 a? And why the data of the growth season 2016-2017 is missing from the middle figure of figure 2a?
  11. Figure 8, the figure was based on the data of which year?

Reviewer 2 Report

Abstract

Unexplained abbreviations in the abstract, for example, CS.

Introduction

The introduction is too long and too general, for example, Lines 31-38 repeat very well-known truths; the same refers to Lines 53-60.

I do not see any necessity for Figure 1. Such a picture can be taken in very different field trials and, in reality, does not show anything specific.

Methods

When and where was the severity of yellow rust scored? How was this estimation related to fungicide application schemes?

What does it mean – 6.1 l. ha-1 fungicides? Table S3 cannot be understood. Firstly, it is a wrong term – “pesticides/fungicides”. Fungicides are a subdivision of the pesticides. It is crucially necessary to show active ingredients, because trade names are different in different countries, I did not understand which type of fungicides was used. As I understand, application of fungicides differed by years. Why? Fungicides should be shown in another table. I did not understand – were insecticides and herbicides used in all three systems or weren’t?

Results

I did not find any results that provide statistically significant reduction in yellow rust severity, because the results fluctuated on a large scale. Of course, it is understandable, because there are many cultivars, possibly with different levels of resistance. Results could be influenced by other pathogens, too.

Please, do not use the term “susceptibility”; actually, you can speak only about the response to yellow rust, as you have correctly written in “Materials and Methods”.

I did not understand Figure 5 – what do the different colours mean?

Discussion

Authors claim that the application of fungicides improves the yield. But – is it so? Perhaps, it depends on cultivar resistance level and the pressure of diseases.

Reviewer 3 Report

I carefully read the submitted manuscript.

This study evaluated the effects of fungicide application to the nitrogen use efficiency, grain yield and quality of winter wheat. The investigation evaluated combination effects of the application of fungicide and different amounts of nitrogen fertilization among three years. I really think data are solid and topic is interesting for readers and suitable for Agronomy.

The manuscript is well constructed and easy to follow the whole story. Also, statistical test were applied appropriately.

 Here, I suggest some minor suggestion.

line 71-72: I think this sentences did not corresponded to Figure 1, so confirmation should be needed.

Figure2A:

I think all of the figure and table should be self-explanatory.

So, the meanings of "*" in figure 2A should be mentioned

line 247:

at the end of the legend, I think ")"was lacking.

That's all.
